# Involvement of enhanced expression of classical complement C1q in atherosclerosis progression and plaque instability: C1q as an indicator of clinical outcome

Shoh Sasaki[1], Kensaku Nishihira[2], Atsushi Yamashita[3], Tomomi Fujii[1], Kenji Onoue[4], Yoshihiko Saito[4], Kinta Hatakeyama[1,5]*, Yoshisato Shibata[2], Yujiro Asada[3], Chiho Ohbayashi[1]

1 Department of Diagnostic Pathology, Nara Medical University, Kashihara, Nara, Japan, 2 Miyazaki Medical Association Hospital, Miyazaki, Japan, 3 Department of Pathology, Faculty of Medicine, University of Miyazaki, Kiyotake, Miyazaki, Japan, 4 Cardiovascular Medicine, Nara Medical University, Kashihara, Nara, Japan, 5 Department of Pathology, National Cerebral and Cardiovascular Center, Suita, Osaka, Japan

* kpathol@naramed-u.ac.jp

**Data Availability Statement:** All relevant data are within the paper and its Supporting information files.

## Abstract

Activation of the classical complement pathway plays a major role in regulating atherosclerosis progression, and it is believed to have both proatherogenic and atheroprotective effects. This study focused on C1q, the first protein in the classical pathway, and examined its potentialities of plaque progression and instability and its relationship with clinical outcomes. To assess the localization and quantity of C1q expression in various stages of atherosclerosis, immunohistochemistry, western blotting, and real-time polymerase chain reaction (PCR) were performed using abdominal aortas from eight autopsy cases. C1q immunoreactivity in relation to plaque instability and clinical outcomes was also examined using directional coronary atherectomy (DCA) samples from 19 patients with acute coronary syndromes (ACS) and 18 patients with stable angina pectoris (SAP) and coronary aspirated specimens from 38 patients with acute myocardial infarction. C1q immunoreactivity was localized in the extracellular matrix, necrotic cores, macrophages and smooth muscle cells in atherosclerotic lesions. Western blotting and real-time PCR illustrated that C1q protein and mRNA expression was significantly higher in advanced lesions than in early lesions. Immunohistochemical analysis using DCA specimens revealed that C1q expression was significantly higher in ACS plaques than in SAP plaques. Finally, immunohistochemical analysis using thrombus aspiration specimens demonstrated that histopathological C1q in aspirated coronary materials could be an indicator of poor medical condition. Our results indicated that C1q is significantly involved in atherosclerosis progression and plaque instability, and it could be considered as one of the indicators of cardiovascular outcomes.

**Funding:** The authors received no specific funding for this work.

**Competing interests:** The authors have declared that no competing interests exist.

# Introduction

Atherosclerosis is an inflammatory disease [1], and its associated complement activation is intimately involved in the development of multiple chronic inflammatory diseases [2, 3]. Some studies suggested that the complement system is activated within atherosclerotic plaques. Although the role of complement in atherogenesis is complicated and not fully understood, its activation has dual effects: proatherogenic and atheroprotective [4]. The classical pathway is activated mainly by immune complexes, microbial and apoptotic cells, and pattern recognition molecules such as C-reactive protein (CRP). CRP is present in the intima of atherosclerotic plaques [5, 6], and autoantibodies against oxidized low-density lipoprotein (OxLDL) [7] and heat shock proteins [8] are also found in atherosclerotic lesions. C1q and mannose-binding lectin can trigger a rapidly enhanced phagocytosis resulting in the clearance of cellular debris, apoptotic cells, and immune complexes, which have atheroprotective effects [9]. Conversely, activation of the complement cascade beyond the C3 convertase with the formation of anaphy-latoxins and the terminal complement complex can induce proinflammatory signaling, which has proatherogenic effects [4]. In this study, we focused on C1q, the first protein in the classical pathway, and assessed its association with atherosclerotic lesions. First, we examined whether C1q protein and mRNA expression is involved in atherosclerosis progression using autopsy specimens of abdominal aortas with various degrees of atherosclerosis. Next, we assessed C1q immunoreactivity in the culprit coronary plaques obtained by directional coronary atherect-omy (DCA) and its association with acute coronary syndromes (ACS). Finally, aspirated coronary materials obtained from patients with acute myocardial infarction (AMI) were investigated to evaluate whether C1q immunoreactivity in the aspirated plaque is related to clinical outcomes.

# Materials and methods

## Specimens

The abdominal aortas of eight patients (cases 1–8) autopsied at Nara Medical University Hospital were examined. Postmortem abdominal aortas were removed as previously described [10]. Nineteen fresh aortic tissues (lesion numbers 1–19) were obtained from 8 abdominal aortas with various degrees of atherosclerosis (S1 Fig, S1 Table). When we extracted several samples from a single case, the samples were taken from separate tissues. Each tissue was cut into two or three specimens. Naturally, these specimens were placed next to each other with similar gross appearance. In one specimen from each tissue, the aortic intimae were mechanically separated from the media and stored at −80˚C until immediately before use for western blot or real-time polymerase chain reaction (PCR) analyses. Another specimen was fixed in 4% para-formaldehyde and embedded in paraffin (S1 Fig, S1 Table). Then, tissue sections were stained with hematoxylin and eosin (HE) using an autostainer (Tissue-Tek Prisma Plus, Sakura Fine-tek Japan Co., Ltd., Tokyo, Japan), and histological observation was performed. Serial sections were then examined immunohistochemically as described in the immunohistochemistry and quantitative methods using image analysis section. In cases 1–4, fresh tissues were snap-frozen in liquid nitrogen, and 4-µm-thick frozen sections of these tissues were stored at −80˚C until use for the immunohistochemistry to confirm the specificity of the primary antibody against C1q (S1 Fig, S1 Table). Atherosclerotic lesions were histologically classified as early and advanced lesions according to the American Heart Association classification [11]. Early lesions were identified as fatty dots or streaks containing some lipid-laden macrophages, corresponding to type I or II lesions. Advanced lesions were considered as atheromatous, calcified,

fibrous, or complicated plaques, corresponding to type IV, V, or VI lesions. Written informed consent was obtained from the families of all autopsies.

Next, culprit coronary plaques obtained from 37 patients with ACS (n = 19) or stable angina pectoris (SAP; n = 18) who underwent DCA at Miyazaki Medical Association Hospital were examined, and coronary aspirated materials obtained from 38 patients with AMI at Miyazaki Medical Association Hospital were also examined. All specimens of DCA samples and aspirated materials were fixed in 4% paraformaldehyde and embedded in paraffin. Then, tissue sections were stained with HE as described previously in the text, followed by histological observation. Immunohistochemical analysis was then performed using serial sections as described in the text. All patients provided written informed consent. This study was approved by the Human Investigation Review Committee of University of Miyazaki (No. 2014–24) and Nara Medical University (No. 2787), and the protocol conformed to the principles outlined in the Declaration of Helsinki (*Cardiovasc Res*. 1997;35:2–4).

## Immunohistochemistry and quantitative methods using image analysis

Approximately 4-μm-thick serial sections were immunohistochemically stained using primary antibodies [12]. Briefly, these sections were deparaffinized, rehydrated with 0.05 mol/L Tris-buffered saline (TBS), and then incubated in 3% hydrogen peroxide in methanol for 20 min to block endogenous peroxidase activity. After washing in TBS, the sections were preincubated with 5% skim milk for 60 min and subsequently incubated with primary antibodies against C1q (clone 9A7; Abcam, Tokyo, Japan), CD68 (clone PGM1; Dako Japan Inc., Kyoto, Japan), or α-smooth muscle actin (clone HHF35; Dako Japan Inc.) for 18 h at 4˚C. Intervening washes in TBS were followed by incubation with the Envision+kit (Dako Japan Inc.) for 30 min at room temperature. After further washes in TBS, horseradish peroxidase activity was visualized using 3, 3′-diaminobenzidine–containing hydrogen peroxide. The sections were counter-stained with Mayer's hematoxylin. The specificity of primary antibody for C1q was examined by immunohistochemistry using frozen sections of aortic atherosclerotic lesions and 4% para-formaldehyde-fixed, paraffin-embedded tissue sections and confirmed using another anti-C1q antibody (clone EPR2981; Abcam). The immunohistochemical staining of frozen sections was performed as described paraffin sections except for the exclusion of deparaffinization. The negative control featured a normal mouse IgG instead of the primary antibody. Immunopositive areas of sections stained with the antibody against C1q were quantified using a color imaging analysis system (Win ROOF, Mitani, Fukui, Japan) and expressed as the percentage of positively stained areas relative to the whole fragmented tissue in the DCA specimen or plaque component in the aspirated coronary material, as previously described [13, 14]. Macrophages (CD68-positive cells) and smooth muscle cells (α-smooth muscle actin-positive cells) were counted (numbers per square millimeter) in the DCA specimen as described previously [6], and the presence or absence of a necrotic core was examined in each case. To identify C1q-positive cells, double staining was performed using BOND-III automated immunohistochemical staining system (Leica Microsystems K. K., Tokyo, Japan) with a BOND Polymer Refine Detection kit (Leica Microsystems K. K.). The chromogenic substrate used in this kit was Fast Red (Leica Microsystems K. K.) or PermaBlue Plus/AP (Diagnostic BioSystems, CA).

## Western blot analysis

Frozen tissues of 10 abdominal aortic intimae including four early lesions and six advanced lesions (S1 Table) were solubilized with lysis buffer (1% Triton X in phosphate buffer saline and 50 mg/mL aprotinin). Then, the quantity of the whole protein was uniformed in each solubilized tissue, using a TaKaRa BCA Protein Assay Kit (bicinchoninate method, a modified

Lowry method) (Takara Bio Inc., Shiga, Japan). Proteins were electrophoresed under reducing conditions on 10%–20% polyacrylamide gels (Wako Pure Chemical Industries Ltd., Osaka, Japan). Separated proteins were transferred to a nitrocellulose membrane. After blocking the nonspecific background with 4% skim milk, the membrane was incubated with a primary antibody against C1q (clone 9A7; Abcam, Tokyo, Japan) or β-actin (clone AC-74; Sigma-Aldrich Corp., St. Louis, MO, USA), followed by incubation with a peroxidase-conjugated secondary antibody (GE Healthcare, Buckinghamshire, UK). Immunoreactive bands were visualized using ECL reagents (GE Healthcare) with X-Omat AR film (Eastman Kodak Co., New York, NY, USA), and band intensities were analyzed using LAS-4000 luminescent image (FujiFilm, Tokyo, Japan) as previously described [15]. C1q expression was normalized to β-actin expression.

### Quantitative real-time PCR analysis

Total RNAs were extracted from nine frozen tissue of abdominal aortic intimae, including five early lesions and four advanced lesions (S1 Table) with TRIzol (Life Technologies Corp., Grand Island, NY, USA) according to the manufacturer's protocols. The initial cDNA strand was synthesized using SuperScript III transcriptase (Life Technologies Corp.) and an oligo-dT primer according to the manufacturer's instructions. Real-time PCR was performed using the Thermal Cycler Dice Real-Time System (Takara Bio Inc., Shiga, Japan). PCR amplification of C1q and β-actin was performed using the following primer pairs: C1q (fwd-5′-gcctcacaggacac-cagctt-3′ and rev-5′-ccaggagcaggagcaacatc-3′) and β-actin (fwd-5′-tggcacccagcacaatgaa-3′ and rev-5′-ctaagtcatagtccgcctagaagca-3′). Quantitative real-time analysis was performed using the QuantiTect SYBR Green PCR Kit (Qiagen Inc., Hilden, Germany) and the following cycling conditions: 95˚C for 15 min, followed by 40 cycles of 94˚C for 15 s, 53˚C for 20 s, and 72˚C for 10 s. Then, a melting curve program was used with continuous fluorescence measurement. A standard curve for C1q templates was drawn through PCR amplification using serial sample dilutions. C1q expression was normalized to β-actin expression [12].

### Statistical analysis

Data are presented as numbers with percentages or means with standard errors of the mean. Quantitative variables between two groups were compared using Student's t-test, Welch's t-test, or the Mann–Whitney U-test as appropriate. Categorical variables were compared using the Fisher's test or the chi-squared test. All tests were two sided and p-values of <0.05 were considered significant.

## Results

### C1q protein and mRNA expression in atherosclerotic lesions in the autopsy specimens

Table 1 displays the clinical characteristics of eight patients, including age, sex, major and concurrent lesions, and the results of the histological evaluation (early or advanced lesions). In western blot analysis, the early lesion group comprises four samples (two each from cases 1 and 2), whereas the advanced lesion group consists of six samples (one each from cases 3 and 4 and two each from cases 7 and 8) (S1 Fig, S1 Table). In real-time PCR analysis, the early lesion group consists of five samples (two from case 1 and three from case 5), whereas the advanced lesion group comprises four samples (one each from cases 3 and 4, and two from case 6) (S1 Fig, S1 Table).

**Table 1. Clinical characteristics and histological evaluation of autopsy cases.**

| Case | Age/Sex | Histology | Major lesion | Concurrent lesion |
|:---:|:---:|---|---|---|
| 1 | 18/M | Early lesion | Pulmonary fibrosis | Acute tubular necrosis |
| 2 | 58/M | Early lesion | Gastrointestinal bleeding | Diabetes mellitus |
| | | | | Cardiac hypertrophy |
| 3 | 76/M | Advanced lesion | Cerebral infarction | No particular findings |
| 4 | 69/M | Advanced lesion | Chronic heart failure | Diabetes mellitus |
| | | | Chronic kidney disease | Emphysema |
| 5 | 68/F | Early lesion | Pulmonary fibrosis | No particular findings |
| 6 | 78/M | Advanced lesion | Emphysema, Cor pulmonale | Abdominal aortic aneurysm |
| 7 | 80/M | Advanced lesion | Acute myocardial infarction | No particular findings |
| 8 | 71/M | Advanced lesion | Deep vein thrombosis | Ulcers in the small intestine and stomach |
| | | | Pulmonary embolism | |

M, male; F, female.

Immunohistochemically, the antibody against C1q stained the extracellular matrix, necrotic cores, macrophages, and smooth muscle cells (Fig 1A–1H, S2A–S2C Fig). Using immunohistochemical double staining, the C1q-positive cells were identified as macrophages and smooth muscle cells in the atherosclerotic intima (Fig 1E–1G). This distribution of C1q protein was identical in frozen sections (S2A–S2C Fig), and similar findings were obtained using another primary antibody (clone EPR2981, S3A–S3E Fig). The localization of C1q immunoreactivity in this study was consistent with that in previous findings. C1q production is essentially restricted to macrophages and immature dendritic cells [16, 17]. After its release, C1q binds to immune complexes, microbial cells, apoptotic cells, and pattern recognition molecules including CRP [1]. C1q immunopositivity in smooth muscle cells might reflect CRP production [18]. Early and advanced atherosclerotic lesions exhibited morphological variation in the numbers of macrophages and smooth muscle cells and the nature of the matrix component; therefore, C1q protein expression was considered to vary among these lesions. Western blotting

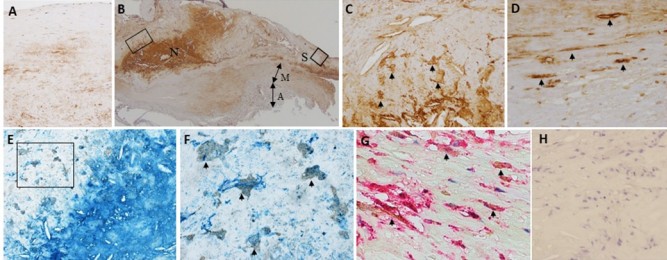

**Fig 1. Representative immunohistochemical results of atherosclerotic lesions in abdominal aortas.** Immunohistochemistry for C1q (A–D). Immunohistochemical double staining of C1q and macrophages (E, F) or smooth muscle cells (G). Extracellular matrix (A), necrotic cores (B), macrophages (C, high-magnification image of the selected area indicated by the rectangle in B), and smooth muscle cells (D, high-magnification image of the selected area indicated by the square in B) were positive for C1q. Arrows denote macrophages and smooth muscle cells that are positive for C1q (C and D, respectively). Middle-magnification image of immunohistochemical double staining of C1q (blue) and macrophages (brown) (E). Many macrophages (brown) are positive for C1q (blue) (F, high-magnification image of the selected area indicated by the rectangle in E), and smooth muscle cells (brown) are also positive for C1q (red) (G). Arrows denote double-positive cells for C1q and macrophages (CD68) (F) or smooth muscle cells (α-smooth muscle actin) (G). Negative control (H). N, necrotic core; S, shoulder region of the plaque; M, media; A, adventitia. A and C, Original magnification ×200; B, Original magnification ×10; D, F–H, Original magnification ×400; E, Original magnification ×100.

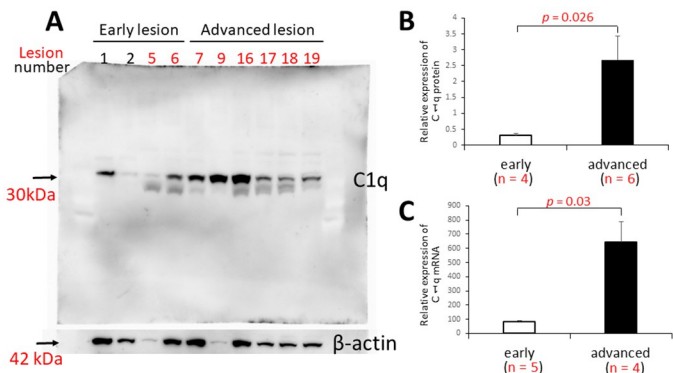

**Fig 2. C1q expression in atherosclerotic lesions in abdominal aortas.** Western blot (A, B) and real-time polymerase chain reaction analysis (C) were performed to evaluate C1q protein and mRNA expression in early and advanced lesions. Early and advanced atherosclerotic lesions exhibited morphological variation in the numbers of macrophages and smooth muscle cells and the nature of the matrix component; therefore, C1q protein expression was considered to vary among these lesions. The relative C1q protein (B) and mRNA (C) expression was significantly higher in advanced lesions than in early lesions ($p = 0.026$ and $p = 0.03$, respectively).

illustrated that C1q protein expression was significantly higher in advanced lesions than in early lesions ($p = 0.026$; Fig 2A and 2B). In addition, real-time PCR demonstrated that the relative mRNA expression of C1q was significantly higher in advanced lesions than in early lesions ($p = 0.03$; Fig 2C).

## C1q expression in the culprit coronary plaques in the DCA specimens

The initial baseline characteristics of 37 patients with SAP or ACS are presented in Table 2. The difference in risk factors for coronary artery disease between the two groups was not significant, excluding the higher proportion of patients prescribed aspirins in the SAP group than in ACS group ($p = 0.05$). The number of macrophages in ACS plaques was significantly higher than that in SAP plaques ($p = 0.009$), but there was no significant difference in the number of smooth muscle cells ($p = 0.14$) or the presence or absence of necrotic cores ($p = 0.50$) between

**Table 2. Clinical characteristics of 37 patients with SAP or ACS.**

| Variable | SAP (n = 18) | ACS (n = 19) | *p* |
|---|---|---|---|
| Age, years | 63 ± 2 | 61 ± 2 | 0.53 |
| Male | 16 (89) | 13 (68) | 0.25 |
| Hypertension | 9 (50) | 13 (68) | 0.58 |
| Hyperlipidemia | 10 (56) | 9 (47) | 0.78 |
| Hyperuricemia | 2 (11) | 8 (42) | 0.21 |
| Diabetes mellitus | 7 (39) | 6 (32) | 0.55 |
| Smoking | 6 (33) | 13 (68) | 0.06 |
| Obesity | 4 (22) | 4 (21) | >0.99 |
| Familial history | 2 (11) | 3 (16) | >0.99 |
| Aspirin | 18 (100) | 13 (68) | 0.05 |
| Statin | 6 (33) | 5 (26) | 0.86 |

Data are expressed as the mean ± standard error of the mean or number (%).

SAP, stable angina pectoris; ACS, acute coronary syndromes.

**Table 3. Comparison of histological characteristics between SAP and ACS.**

|  | SAP (*n* = 18) | ACS (*n* = 19) | *p* |
|---|---|---|---|
| Macrophages (number per mm2) | 15.1 ± 4.3 | 52.4 ± 12.4 | 0.009 |
| Smooth muscle cells (number per mm2) | 122 ± 8.2 | 100 ± 12.4 | 0.14 |
| Necrotic cores | 5 (28) | 8 (42) | 0.50 |

Data are expressed as the mean ± standard error of the mean or number (%).

SAP, stable angina pectoris; ACS, acute coronary syndromes.

the two groups (Table 3). Immunohistochemical analysis illustrated that C1q expression was significantly higher in ACS plaques than in SAP plaques (*p* = 0.034; Fig 3A–3C).

## C1q expression in the aspirated coronary materials of patients with AMI

The investigation using coronary aspirated materials included 188 patients with AMI who underwent primary percutaneous coronary intervention (PCI) within 24 h of symptom onset at Miyazaki Association Hospital between January 2010 and November 2011. Microscopically, the aspirated coronary materials from these patients contained both the thrombus and plaque components (Fig 4A and 4C). Of these 188 patients, 150 patients whose coronary aspirated materials contained little or no plaque components (<1 mm$^2$) were excluded. Ultimately, 38 patients with AMI were enrolled in the study. The clinical and angiographic characteristics of the remaining 38 patients are presented in Table 4. The areas of immunopositivity for C1q were quantified using a color imaging analysis system as described in the methods section [13, 14], and various degree of positive staining were observed in the plaque components of the aspirated coronary material (Fig 4B and 4D), whereas the thrombus component was invariably immunonegative (Fig 4B and 4D).

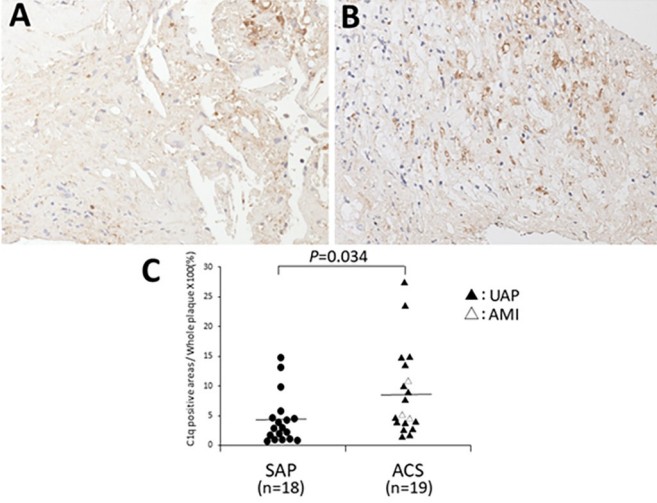

**Fig 3. Representative microphotographs of the C1q expression in the culprit coronary plaques in DCA specimens.** Immunostaining for C1q was performed in the SAP (A) and ACS (B) plaques. The percentage of C1q-positive areas relative to the whole fragmented tissue was significantly higher in ACS plaques than in SAP plaques (*p* = 0.034). Each symbol represents the ratio of C1q-positive areas to the whole fragmented tissue of one independent case. The horizontal bar represents the mean value (C). DCA, directional coronary atherectomy; SAP, stable angina pectoris; ACS, acute coronary syndromes; UAP, unstable angina pectoris; AMI, acute myocardial infarction. A and B, original magnification ×200.

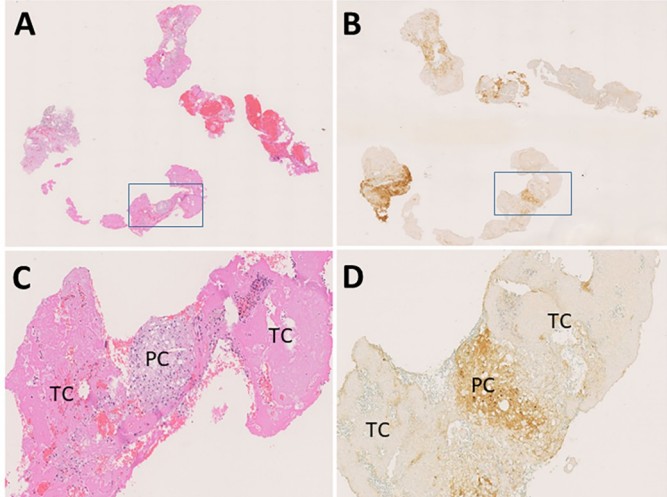

**Fig 4. Representative microphotographs of the aspirated coronary materials from patients in the high C1q expression group.** A-B or C-D are serial sections, respectively. The aspirated coronary materials contain the plaque and thrombus component (A) and are immunopositive for C1q to some extent (B). High magnification images of the selected areas are indicated by the rectangles in A or B. The plaque component is strongly immunopositive for C1q, whereas the thrombus component is completely immunonegative (C, D). PC, plaque component; TC, thrombus component. A and B, original magnification ×20; C and D, original magnification ×200.

**Table 4. Clinical and angiographic characteristics of 38 patients with acute myocardial infarction.**

| Variable | Low C1q expression (n = 15) | High C1q expression (n = 23) | *p* |
|---|---|---|---|
| Age, years | 69 ± 3 | 68 ± 3 | 0.84 |
| Male | 12 (80) | 16 (70) | 0.71 |
| Hypertension | 8 (53) | 14 (61) | 0.65 |
| Hyperlipidemia | 5 (33) | 14 (61) | 0.1 |
| Diabetes mellitus | 5 (33) | 9 (39) | 0.72 |
| Renal insufficiency | 6 (40) | 8 (35) | 0.74 |
| Statin | 1 (7) | 4 (17) | 0.15 |
| HDL cholesterol, mg/dL | 48.5 ± 3.3 | 48 ± 2.3 | 0.91 |
| LDL cholesterol, mg/dL | 116.1 ± 6.1 | 143.3 ± 8.5 | 0.013 |
| Triglycerides, mg/dL | 83.2 ± 14 | 94.7 ± 10.5 | 0.51 |
| Hs-CRP, mg/dL | 0.4 ± 0.12 | 0.21 ± 0.06 | 0.16 |
| Peak CK, IU/L | 4342 ± 947 | 4298 ± 837 | 0.97 |
| STEMI | 15 (100) | 21 (91) | 0.51 |
| Onset-to-reperfusion time < 12 h | 14 (93) | 19 (83) | 0.63 |
| Multivessel disease | 10 (67) | 11 (48) | 0.25 |
| Length of hospital stays, days | 15.3 ± 1.3 | 19.1 ± 1.7 | 0.08 |
| MACCE at 1 year after PCI | 2 (13) | 6 (26) | 0.44 |

Data are expressed as the mean ± standard error of the mean or number (%).

HDL, high-density lipoprotein; LDL, low-density lipoprotein; Hs-CRP, high-sensitivity C-reactive protein; CK, creatine kinase; STEMI, ST-elevated myocardial infarction; MACCE, major adverse cardiac and cerebrovascular event; PCI, percutaneous coronary intervention. Renal insufficiency was defined as eGFR < 60 mL/min/ 1.73 m$^2$.

Each patient with AMI was examined for C1q immunopositivity in the plaque component. Because the average proportion of C1q-positive tissue in DCA samples from patients with ACS was approximately 10%, 10% was used as the cut-off for aspirated tissue from AMI patients. The 38 patients with AMI were divided into high (n = 23) and low (n = 15) C1q expression groups using aforementioned threshold. We evaluated the relationship between C1q protein expression and clinical outcomes, including the length of hospital stay and occurrence of major adverse cardiac and cerebrovascular events (MACCEs) defined as all-cause death, stroke, or myocardial infarction (MI) within 1 year of PCI (Table 4). MACCE occurred in six (26%) patients in the high C1q expression group (three deaths, two strokes, and one MI) and two (13%) patients in the low C1q expression group (one death and one stroke). The length of hospital stays tended to be longer in the high C1q expression group ($p$ = 0.08). Additionally, serum low-density lipoprotein (LDL) levels were significantly higher in the high C1q expression group than in the low C1q expression group ($p$ = 0.013).

## Discussion

The major findings of this study were that the relative protein and mRNA expression of C1q was significantly higher in advanced atherosclerotic plaques than in early lesions and that C1q expression was significantly higher in the culprit plaques of patients with ACS than in those of patients with SAP. Furthermore, C1q immunoreactivity levels in the aspirated coronary materials from patients with AMI were significantly associated with serum LDL levels, and they tended to correlate with the length of hospital stay.

Analysis using autopsy specimens indicated that C1q protein and mRNA expression was significantly higher in advanced lesions than in early lesions. These results indicated that C1q is involved in atherosclerosis progression, and C1q expression may be considered a good indicator of the degree of atherosclerosis progression. Anaphylatoxins and the terminal complement complex, defined as a downstream protein of the complement pathway, promote atherosclerosis progression via multiple mechanisms, such as inflammatory cytokine and monocyte chemoattractant protein-1 production, adhesion molecule expression, cell proliferation, and growth factor release [4]. Conversely, C1q is an important protein for efferocytosis [9], and therefore, C1q has both proatherogenic and atheroprotective effects. However, Schrijvers et al. reported that efferocytosis is impaired in advanced atherosclerotic plaques attributable to several factors, such as oxidative stress, the accumulation of indigestible materials in the macrophage cytoplasm, and the presence of OxLDL or oxidized red blood cells [19]. Thus, the actual atheroprotective effect of C1q appears to be weaker in advanced lesions than in early lesions.

Immunohistochemical analysis using DCA specimens indicated that C1q protein expression was significantly higher in ACS plaques than in SAP plaques. This finding may reflect that CRP and OxLDL are abundantly found in ACS plaques. CRP and OxLDL are important activators of the classical pathway and it has been demonstrated that quantities of these proteins are significantly or tend to be larger in patients with unstable angina and AMI than in those with stable angina according to histological and serological studies [6, 20–25]. In addition, C5a has been demonstrated to induce matrix metalloproteinase-1 (MMP-1) and MMP-9 expressions in human macrophages, which results in plaque instability [26]. Although it is unclear how much C1q is actually derived from lesional macrophages or blood monocytes in the context of ACS, the numbers of macrophages were significantly higher in ACS plaques than in SAP plaques in the present study, suggesting that lesional macrophages can affect the expression and deposition of C1q in ACS plaques. Tran et al. revealed that C1q expression is significantly upregulated by the transcription factor MafB, which is expressed selectively by

monocytes and macrophages [27]. Several studies also demonstrated that MafB has an anti-apoptotic role [28–30], and Hamada et al. found that MafB participates through the expression of apoptosis inhibitor of macrophages (AIM) [29]. In advanced lesions, macrophage apoptosis promotes lesion vulnerability and necrotic core development via defective efferocytosis, resulting in plaque instability [31]. Taken together, these findings suggest that MafB significantly regulates homeostasis in atherosclerotic lesions through C1q and AIM expression, and that C1q plays an important role in plaque instability. Concerning the function of C1q protein in the onset of AMI, interactions between complement components and coagulation factors or platelets may be speculated, especially in the process of thrombus formation at the plaque disruption site of the coronary artery [32, 33]. This would be an important aspect of C1q function; therefore, further studies are needed to elucidate the role of the complement system in the regulation of thrombus initiation and propagation.

Immunohistochemical analysis using thrombus aspiration specimens demonstrated that histopathological C1q levels were positively correlated with serum LDL levels. This result was expected, because serum circulating OxLDL levels were significantly correlated with serum total LDL levels [34]; thus, OxLDL can activate the classical pathway, and it appears to affect CRP-induced complement activation [35, 36]. Conversely, no significant difference was observed in high-sensitivity CRP (hs-CRP) levels between the two groups, contradicting our expectation. This finding suggests that hs-CRP levels in patients with AMI might reflect immune activation related to the proatherogenic process, inflammatory mechanisms that cause acute coronary events, and inflammatory responses associated with the presence of necrotic myocardial cells. Although no significant difference was observed in the MACCE rate between the two groups, the length of hospital stays tended to be longer in the high C1q expression group, indicating that histopathological C1q expression in the aspirated coronary material could be a prognostic factor.

This study had several limitations. First, the potential role of the classical pathway in atherosclerotic lesions compared with that of the other complement pathways was not evaluated, even though we focused on C1q, the first protein in the classical pathway. Moreover, our results, together with those of previous reports, suggest that C1q significantly contributes to atherosclerosis progression and especially plaque instability. Further investigations evaluating multiple complement proteins in addition to C1q are required to resolve this issue. Second, MafB expression and its correlation with C1q expression were not evaluated. However, the number of macrophages included in each sample probably affected the result; therefore, evaluating the accurate correlation between MafB and C1q expression would be difficult using plaque specimens. Third, although all findings for each examination in this study indicated that C1q is involved in atherosclerosis progression and the mechanisms that lead to plaque rupture, the number of patients was small. In particular, the potential correlation of histopathological C1q expression with clinical outcomes should be investigated further in larger studies. Moreover, additional objective data such as carotid intima-media thickness and plaque burden are necessary to examine whether C1q expression can be an indicator of atherosclerotic progression and a predictor of cardiovascular disease. Finally, the pathophysiological explanations for our findings are speculative and the underlying mechanisms remain unclear.

## Conclusions

In conclusion, we demonstrated that C1q has potential roles in atherosclerosis progression and plaque instability and it could be considered as one of the indicators of cardiovascular outcomes.

## Supporting information

**S1 Fig. Sampling method of aortic tissues (case 1) for morphological study (HE stain and IHC), western blot or real time-PCR.** Each lesion tissue was cut into two or three specimens. One specimen was fixed in 4%PFA and embedded in the paraffin, followed by HE stain and IHC. The aortic intimae of another specimen were mechanically separated from the media and stored at -80˚C until immediately before use for western blot or real-time PCR analyses. In lesion 1, the other specimen was snap-frozen in liquid nitrogen, and serial frozen sections was stored at -80˚C until use for IHC. 4%PFA, 4% paraformaldehyde; IHC, immunohisto-chemical stain; PCR, polymerase chain reaction.
(TIF)

**S2 Fig. Immunohistochemical results of advanced atherosclerotic lesions (frozen sections).** A, B and C are serial sections. Many cells are immunopositive for C1q in atheromatous plaque (A). C1q immunoreactivity is found in many macrophages (B, CD68) and some smooth muscle cells (C, α-smooth muscle actin). Arrows and arrowheads show C1q-positive macrophages and smooth muscle cell, respectively. A-C, original magnification x100.
(TIF)

**S3 Fig.** Immunohistochemistry for C1q using another anti-C1q antibody (clone EPR2981) (A). Immunohistochemical double staining of C1q (clone EPR2981) and macrophages (CD68, B and C) or smooth muscle cells (α-smooth muscle actin, D) (B–D). A and B are serial sections. Necrotic core, extracellular matrix, and many cells were immunopositive for C1q in atheromatous plaques (A). Middle-magnification image of immunohistochemical double staining of C1q (blue) and macrophages (brown) (B). Many macrophages (brown) are positive for C1q (blue) (C, high-magnification image of the selected area indicated by the rectangle in B), and smooth muscle cells (brown) are also positive for C1q (red) (D). Arrows denote double-positive cells for C1q and macrophages (C) or smooth muscle cells (D). Negative control (E). A, original magnification ×40; B, Original magnification ×100; C–E, Original magnification ×200.
(TIF)

**S1 Table.**
(DOCX)

**S1 Raw images.**
(TIF)

## Author Contributions

**Conceptualization:** Kinta Hatakeyama, Yujiro Asada.

**Data curation:** Shoh Sasaki, Atsushi Yamashita, Kenji Onoue, Kinta Hatakeyama, Yoshisato Shibata.

**Formal analysis:** Kinta Hatakeyama.

**Investigation:** Shoh Sasaki, Kensaku Nishihira, Atsushi Yamashita, Tomomi Fujii, Kenji Onoue, Yoshihiko Saito, Kinta Hatakeyama, Yoshisato Shibata, Yujiro Asada, Chiho Ohbayashi.

**Methodology:** Shoh Sasaki, Kensaku Nishihira, Atsushi Yamashita, Tomomi Fujii, Kenji Onoue, Yoshihiko Saito, Kinta Hatakeyama, Yoshisato Shibata, Yujiro Asada, Chiho Ohbayashi.

**Resources:** Kinta Hatakeyama, Yujiro Asada.

**Software:** Kinta Hatakeyama.

**Supervision:** Kensaku Nishihira, Kinta Hatakeyama.

**Validation:** Kensaku Nishihira, Atsushi Yamashita, Kinta Hatakeyama.

**Visualization:** Kinta Hatakeyama.

**Writing – original draft:** Shoh Sasaki.

**Writing – review & editing:** Kinta Hatakeyama.

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
