## [Decision Letter · Decision Letter 0]

22 Feb 2021

PONE-D-21-00264

Involvement of enhanced expression of classical complement C1q in the atherosclerosis progression and plaque instability: C1q as an indicator of clinical outcome

PLOS ONE

Dear Dr. Hatakeyama,

Thank you for submitting your manuscript to PLOS ONE. After careful consideration, we feel that it has merit but does not fully meet PLOS ONE’s publication criteria as it currently stands. Therefore, we invite you to submit a revised version of the manuscript that addresses the points raised during the review process.

We look forward to receiving your revised manuscript.

Kind regards,

Christoph E Hagemeyer, PhD

Academic Editor

PLOS ONE

Journal Requirements:

2)  PLOS ONE now requires that authors provide the original uncropped and unadjusted images underlying all blot or gel results reported in a submission’s figures or Supporting Information files. This policy and the journal’s other requirements for blot/gel reporting and figure preparation are described in detail at https://journals.plos.org/plosone/s/figures#loc-blot-and-gel-reporting-requirements and https://journals.plos.org/plosone/s/figures#loc-preparing-figures-from-image-files. When you submit your revised manuscript, please ensure that your figures adhere fully to these guidelines and provide the original underlying images for all blot or gel data reported in your submission. See the following link for instructions on providing the original image data: https://journals.plos.org/plosone/s/figures#loc-original-images-for-blots-and-gels.

Reviewers' comments:

Reviewer's Responses to Questions

**Comments to the Author**

1. Is the manuscript technically sound, and do the data support the conclusions?

Reviewer #1: Partly

2. Has the statistical analysis been performed appropriately and rigorously? 

Reviewer #1: Yes

3. Have the authors made all data underlying the findings in their manuscript fully available?

Reviewer #1: Yes

4. Is the manuscript presented in an intelligible fashion and written in standard English?

Reviewer #1: Yes

5. Review Comments to the Author

Reviewer #1: Shoh Sasaki et al reported implication of complement early protein C1q in human atherosclerotic lesions and their potential importance in plaque progression and possibly in plaque ruptures in their work (Involvement of enhanced expression of classical complement C1q in the atherosclerosis progression and plaque instability: C1q as an indicator of clinical outcome). In their experiments, various human atherosclerotic samples were tested to determine protein and mRNA C1q expression using immunohistochemical method, western blot and RT-PCR method. Based on their results, authors concluded that early complement protein, C1q, promotes atherosclerosis progression leading to plaque instability and perhaps subsequent rupture.

Major comments

(1) Results in the manuscript were mainly C1q expressions in different atherosclerotic lesions from different group. As per conclusion of “C1q is involved in plaque progression and plaque instability”, I believe authors need to provide missing links i.e. how does it? Assuming all happen in local lesions, first one is how C1q is activated- via antibodies or other protein etc., next is how they promote vulnerable plaques – via efferocytosis or inflammatory cytokines or target cell death. Is this more relevant to see link with oxLDL antibodies than oxLDL? (Authors measured only lipid profile though!). Exploring these will help to strength the authors’ conclusion.

(2) Most immunohistochemical stainings were done using sequential sections. It will be better to visualise the results in double-immunostaining with magnified views. Figure 1 and supplementary figures 2 and 3. This will definitely help to confirm co-expression of C1q and respective target cells.

(3) I am not sure how samples were distributed in Western Blot and RT-PCR. It seems to me that two different samples were used from some autopsied tissues (Page 11). For example, samples in Western blot were 1, 2, 3, and 4 assuming that 1 and 2 were from case 1 and 3 and 4 were from case 2. However 1 and 2 were so much different in protein expression. Please confirm whether these samples are in same classifications.

(4) Figure Legend in Figure 3 states that immunostaining per whole plaques. Can you please show the low-power images, i.e. whole plaques?

(5) In ruptured plaques, necrotic core is a hallmark feature. In addition to lipid cores (lipid accumulation), it is better to have necrotic core (cell deaths).

(6) I am not clear about how C1q content in aspirated coronary materials was measured and the reason of using 10% cut-off level was used to categorise low and high C1q expression.

Minor comments

(1) Please clarify following staining procedure, i.e. whether “tissue sections were stained with hematoxylin and eosin (HE) first and then with primary antibodies” or “tissue sections were stained with primary antibodies and counterstained with H&E”.

(2) It is recommended to have better counterstain to visualise cells.

(3) In Specimens (Materials and Methods), it states that all samples were prepared and embedded in paraffin. However frozen sections were used (in page 8 and in supplementary figure 2 pages 12 and 31)

6. PLOS authors have the option to publish the peer review history of their article (what does this mean?). If published, this will include your full peer review and any attached files.

Reviewer #1: No

---

## [Author Response · Author response to Decision Letter 0]

6 Aug 2021

Dear Editors

Thank you for inviting us to submit a revised draft of our manuscript to PLOS ONE. We are happy because we get insightful opinions to strengthen our paper. We have incorporated changes that reflect the detailed suggestions you have graciously provided. In the manuscript, the writing in red is the corrected places.

Response to Reviewer #1 :

We greatly appreciate your interest in our paper and valuable comments. 

Major comments

(1) Results in the manuscript were mainly C1q expressions in different atherosclerotic lesions from different group. As per conclusion of “C1q is involved in plaque progression and plaque instability”, I believe authors need to provide missing links i.e. how does it? Assuming all happen in local lesions, first one is how C1q is activated- via antibodies or other protein etc., next is how they promote vulnerable plaques – via efferocytosis or inflammatory cytokines or target cell death. Is this more relevant to see link with oxLDL antibodies than oxLDL? (Authors measured only lipid profile though!). Exploring these will help to strength the authors’ conclusion.

Reply: We are grateful to the reviewer for this important comment. We agree with the reviewer's comments, but unfortunately it is difficult to show the mechanism in this observational study using human samples. In addition, anti-ox-LDL antibodies and blood samples were not available, and the relationship between ox-LDL/ox-LDL antibodies in plaques and C1q is unknown. Based on the reviewers' comments, the above point was added to the limitation section (Page 26, line 398-399)..

(2) Most immunohistochemical stainings were done using sequential sections. It will be better to visualise the results in double-immunostaining with magnified views. Figure 1 and supplementary figures 2 and 3. This will definitely help to confirm co-expression of C1q and respective target cells.

Reply: This is a very important comment. As the reviewer pointed out, we performed double-immunostaining to confirm the types (macrophages or smooth muscle cells) of C1q-positive cells. Because macrophages show autofluorescence in human atheromatous plaque, we performed immunohistochemical double staining with high magnified views instead of the fluorescent study. 

 Microphotographs of immunohistochemical double staining to confirm C1q positive-macrophages (macrophage, brown vs. C1q, blue) and C1q positive-smooth muscle cells (smooth muscle cells, brown vs. C1q, red) were shown in figure 1E-G and supplemental figure 3(B– D). The methods and results of the immunohistochemical double staining were described in the text (Page 9, line 153-157 and Page 13, line 214-216, respectively). 

(3) I am not sure how samples were distributed in Western Blot and RT-PCR. It seems to me that two different samples were used from some autopsied tissues (Page 11). For example, samples in Western blot were 1, 2, 3, and 4 assuming that 1 and 2 were from case 1 and 3 and 4 were from case 2. However 1 and 2 were so much different in protein expression. Please confirm whether these samples are in same classifications.

Reply: We are sorry for the inadequate explanation of methods. We added supplemental figure 1 and supplemental table 1. Supplemental figure 1 showed the details of sampling method of aortic tissue from autopsy cases, and supplemental table 1 showed information of lesion numbers (1-19), case numbers (1-8) and atherosclerotic lesion types (AHA type II-VI). We indicated how 19 atherosclerotic lesions were sampled from 8 autopsy cases and used in western blot (10 lesions) or real time-PCR (9 lesions) in the text (Page 6, line 96-104), supplemental figure 1, and supplemetal table 1. 

Each lesions (lesion number 1-6, 11-13) of early atherosclerosis (AHA type II) show morphological and biological variation in macrophage and the nature of the matrix component, therefore, C1q protein expression is considered to vary among these lesions. We added this description both in the result (Page 14, line 223-225) and in figure 2 legend. 

(4) Figure Legend in Figure 3 states that immunostaining per whole plaques. Can you please show the low-power images, i.e. whole plaques?

Reply: We would like to thank the reviewer for an important comment. We are sorry for writing the incorrect description, “----- positively stained areas per the whole plaque ---”. 

Because DCA specimens are composed of highly fragmented tissues which are sampled from coronary atherosclerotic intima, whole plaques cannot be observed in DCA samples. Therefore, we changes the description of the method (Immunohistochemistry and quantitative methods using the image analysis sectoin) (Page 9, line 146-150) and figure 3 legend. 

(5) In ruptured plaques, necrotic core is a hallmark feature. In addition to lipid cores (lipid accumulation), it is better to have necrotic core (cell deaths).

Reply: We are grateful to the reviewer for the valuable comment. We are sorry for the incorrect description, “lipid core” in Fig. 1B, Fig. 3A and supplemental Fig. 1A-C in original manuscript. Autopsied specimens of advanced atherosclerotic plaques in this study have necrotic core with cholesterol clefts and degenerated macrophages, and we immunohistochemically examined these advanced lesions with necrotic core. 

Therefore, we changed “lipid core” to “necrotic core” in the text of results, table 3, and figure legends of Fig. 1B and supplemental Fig. 3A.

(6) I am not clear about how C1q content in aspirated coronary materials was measured and the reason of using 10% cut-off level was used to categorise low and high C1q expression.

Reply: We appreciate the reviewer for the comment. We are sorry for the indequate description of measurement of C1q immunopositive areas in the aspirated coronary materials. As described on Page 9, line 146-150 and Page 18, line 290-293, the percentage of C1q-positive areas relative to plaque component was shown in the aspirated thrombus sample. We changed the description of the method about the measurement of positive area ratio a little more detailed (Page 9, line 146-150 and Page 18, line 290-293), and added the reference 14 (Nishihira K, Yamashita A, Imamura T, Hatakeyama K, Sato Y, Nakamura H, Yodoi J, Ogawa H, Kitamura K, Asada Y. Thioredoxin in coronary culprit lesions: possible relationship to oxidative stress and intraplaque hemorrhage. Atherosclerosis. 201:360-367, 2008) in addition to ref 13. 

The reason for using the 10% cut-off level is that, as shown in Fig.3, the average C1q positive area ratio of ACS patients in DCA samples was about 10%. Therefore, 10% was used as the cut-off value for AMI aspirated tissue. We added the sentence of the reason for using the 10% cut-off level in the result section (Page 21, line 313-314). 

Minor comments

(1) Please clarify following staining procedure, i.e. whether “tissue sections were stained with hematoxylin and eosin (HE) first and then with primary antibodies” or “tissue sections were stained with primary antibodies and counterstained with H&E”.

Reply: We would like to thank the reviewer for the comment. We changed the text to improve the explanation of the methods of HE stain (Page 6, 104-107, Page 7, line 121-124) and immunohistochemical stain (Page 8, line 132-139). The method of nuclear counterstain after immunohistochemistry is also indicated in the the method section (Page 8, line 140-141). 

(2) It is recommended to have better counterstain to visualise cells.

Reply: We appreciate the reviewer for the recommendation. To visualize cells, we performed immunohistochemical double stain with counterstain of the nuclei, as indicated in Fig. 1. 

(3) In Specimens (Materials and Methods), it states that all samples were prepared and embedded in paraffin. However frozen sections were used (in page 8 and in supplementary figure 2 pages 12 and 31)

Reply: We are sorry for the inadequate description of materials and methods. We performed the immunostaining of C1q for the confirmation of the specificity of the C1q primary antibody using not only the 4%PFA fixed, paraffin embedded sections but also the frozen sections. We added the sentence of sampling method of frozen tissues and frozen sections (Page 7, line 108-110) in the materials and methods. In addition the method of immunohistochemical stain using frozen sections were shown in the method ( "Immunohistochemistry and quantitative methods using the image analysis") (Page 9, line 144-145).

---

## [Editor Report · Decision Letter 1]

23 Dec 2021

Involvement of enhanced expression of classical complement C1q in the atherosclerosis progression and plaque instability: C1q as an indicator of clinical outcome

PONE-D-21-00264R1

Dear Dr. Hatakeyama,

We’re pleased to inform you that your manuscript has been judged scientifically suitable for publication and will be formally accepted for publication once it meets all outstanding technical requirements.

Kind regards,

Christoph E Hagemeyer, PhD

Academic Editor

PLOS ONE
---

## [Editor Report · Acceptance letter]

4 Jan 2022

PONE-D-21-00264R1 

Involvement of enhanced expression of classical complement C1q in atherosclerosis progression and plaque instability: C1q as an indicator of clinical outcome 

Dear Dr. Hatakeyama:

I'm pleased to inform you that your manuscript has been deemed suitable for publication in PLOS ONE. Congratulations! Your manuscript is now with our production department. 

Kind regards, 

on behalf of

Dr. Christoph E Hagemeyer 

Academic Editor

PLOS ONE